# LAU: A NOVEL TWO-PARAMETER LEARNABLE LOGMOID ACTIVATION UNIT

## ABSTRACT

In this work, we proposed a novel learnable Logmoid Activation Unit (LAU), $f(x) = x \ln(1 + \alpha\mathrm{sigmoid}(\beta x))$ by parameterizing Logmoid with two hyper-parameters $\alpha$ and $\beta$ that are optimized via back-propagation algorithm. We design quasi-interpolation neural network operators with Logmoid-1 for approximating any continuous function in closed spaces. Our simulations show end-to-end learning deep neural networks with learnable Logmoids increase the predictive performances beyond all well-known activation functions for different tasks.

## 1 INTRODUCTION

In recent years, deep learning has achieved remarkable success in various classification problems (LeCun et al., 2015; Sriperumbudur et al., 2010). The main reason is the powerful abilities of deep neural networks (DNNs) in representation and learning for the unknown structures. One of most important components in DNNs is activation function. A well-designed activation function can greatly improve the predictive performances (Krizhevsky et al., 2017). This has intrigued growing interests in exploring activation functions (Nwankpa et al., 2018; Liang & Srikant, 2016; Shen et al., 2019).

Activation functions are generally classified into linear, nonlinear monotonic and nonlinear non-monotonic functions. Although linear functions including Step Function (Klein et al., 2009), Sign Function (Huang & Babri, 1998) and Identity Function have been widely used in early results, they are useless in practical applications because of the discontinuous derivatives, or lack of biological motivation and classification ability. These problems are further addressed by using nonlinear monotonic functions such as Sigmoid, Tanh and ReLU families. While small derivatives of Sigmoid (Hassell et al., 1977) and Tanh (Kalman & Kwasny, 1992) may cause the gradient to disappear (He & Xu, 2010; Klambauer et al., 2017). Softplus and ReLU functions are designed for solving this problem. Meanwhile, ReLU shows other characters such as the educed saturation, sparsity, efficiency, and ease of use, but the neural network may lose some valid information because of all negative values of ReLU being zero. This intrigues designing new activation functions such as Leaky ReLU (Maas et al., 2013), RReLU (Xu et al., 2015), ELU (Clevert et al., 2015) and Swish (Ramachandran et al., 2017).

In general, designing good activation function is still an open question. One method is to find new activation functions by combining different units, such as Mish (Misra, 2019) and TanhExp (Liu & Di, 2020). The second is to parameterize some well-known activation functions Biswas et al. (2020); Zhou et al. (2020), which may show better performance beyond parameter-free functions. One example is Logish Zhu et al. (2021) which exhibits better performance and Top-1 accuracy. Another is Padé Activation Unit (PAU) Molina et al. (2019) using learnable hyper-parameter(s) and back-propagation algorithm (LeCun et al., 1989).

In this work, we proposed a new family of activation functions by parameterizing Logmoid with fewer trainable parameters for each network layer. It is given by

$$f(x; \alpha, \beta) = x \ln(1 + \alpha\mathrm{Sigmoid}(\beta x)) \tag{1}$$

where the logarithmic operation can reduce the range of Sigmoid, and $\alpha$ and $\beta$ are trainable parameters. The main contributions are summarized as follows:

    1. A new family of activation functions called Logmoid family is proposed.

Table 1: The Top-1 accuracy of Logmoid with parameters on CIFAR10.

| Parameter($\alpha = 1$) | Top-1 | Parameter($\beta = 1$) | Top-1 |
|---|---|---|---|
| 10 | 76.93% | 10 | 68.48% |
| 5 | •**78.2%** | 5 | 73.99% |
| 1 | 77.5% | 1 | •**77.5%** |
| 0.01 | 36.2% | 0.01 | - |
| -0.01 | 41.28% | -0.01 | - |
| -1 | 77.11% | -1 | - |
| -10 | 77.06% | -10 | - |

2. Two quasi-interpolation type operators with Logmoid-1 are constructed to approximate univariate continuous functions.

3. The trainable Logmoid Activation Unit performs better than others in simulations.

The rest is organized as follows: Section 2 briefly describes some relative activation functions. Section 3 proposes new activation functions by parameterizing Logmoid. Section 4 analyzes the performances of the subfamily Lgomoid-1. In Section 5, we introduce a learnable activation unit called LAUs, and implement several simulations to verify its effectiveness, while the last section concludes the results.

## 2 RELATED WORKS

This section devotes some related activation functions including Swish, TanhSoft and PAU.

**Swish**– Swish (Ramachandran et al., 2017) is defined by

$$f(x) = x\text{Sigmoid}(x). \tag{2}$$

This function shows new features of unsaturation, smooth and nonmonotonicity. It can potentially present problems in gradient-based optimization because of not continuously differentiable.

**TanhSoft**–TanhSoft family is indexed by four hyper-parameters as

$$f(x; \alpha, \beta, \gamma, \delta) = \tanh(\alpha x + \beta e^{\gamma x}) \ln(\delta + e^x). \tag{3}$$

Based on simulations on CIFAR10 dataset, two sub-families of TanhSoft-1 and TanhSoft-2 are defined as

$$\text{TanhSoft-1}: \quad f(x; \alpha, 0, \gamma, 1) = \tanh(\alpha x) \ln(1 + e^x), \tag{4}$$

$$\text{TanhSoft-2}: \quad f(x; 0, \beta, \gamma, 0) = x \tanh(\beta e^{\gamma x}). \tag{5}$$

TanhSoft-1 ($\alpha = 0.87$) and TanhSoft-2 ($\beta = 0.6, \gamma = 1$) may show higher Top-1 performance than others Biswas et al. (2020).

**PAU**–Padé Activation Units (PAU) is a learnable activation function based on rational function. The padé is a rational function of the form

$$F(x) = \frac{a_0 + a_1 x + a_2 x^2 + \cdots + a_m x^m}{1 + b_1 x + b_2 x^2 + \cdots + b_n x^n} \tag{6}$$

PAU allows free parameters which can be optimized end-to-end in neural networks.

## 3 LOGMOID ACTIVATION FUNCTION FAMILY

In this section we present a new family of activation function by parameterizing Logmoid with two hyper-parameters. Specially, we propose a family of functions with two hyper-parameters as

$$f(x; \alpha, \beta) = x \ln(1 + \alpha\sigma(\beta x)) \tag{7}$$

where $\sigma$ refers to Sigmoid $\sigma(x) = \frac{1}{1+e^{-x}}$. Table 1 shows its Top-1 accuracy. For $\alpha = 1$, the accuracy decreases faster for $-1 < \beta < 1$, and reaches to 78.2% for $\beta = 5$. For $\beta = 1$, it decreases faster with the gradual increase of $\alpha$. This means Logmoid with $\alpha = 1$ has a better performance.

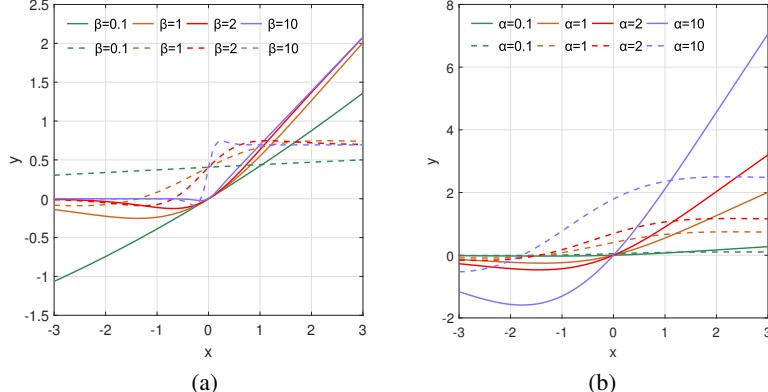

Fig. 1: (a) Numerical Logmoid (solid line) and its derivative (dashed) in terms of $\beta$ and $\alpha = 1$. (b) Numerical Logmoid (solid line) and its first derivative (dashed line) with respect to $\alpha$ and $\beta = 1$.

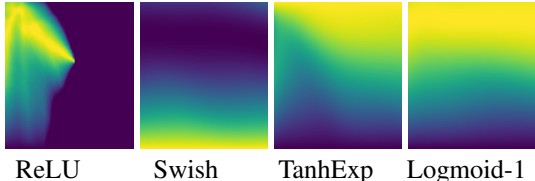

ReLU      Swish     TanhExp   Logmoid-1

Fig. 2: Comparison between the output landscapes of ReLU, Swish, TanhExp and Logmoid-1 on a 5-layer network.

Fig.1(a) shows Logmoid and its derivative in terms of $\beta$. For $\alpha = 1$, Logmoid is a scaled linear function $f(x) = kx$ if $\beta \to 0$. Logmoid is then regarded as an interpolation between linear function and ReLU. Here, $\beta$ shows how fast the first derivative changes asymptotically. When $\beta = 1$, it keeps the gradient property of ReLU for solving the dying problem. From Fig.1(b), Logmoid is far away from the linear function $f(x) = x$ in the positive part of $x$-axis for large $\alpha$ and $\beta = 1$. For practical applications, one may choose hyper-parameters from $1 \leqslant \alpha \leqslant 5$ and $1 \leqslant \beta \leqslant 5$. This is further regarded as sub-family Logmoid-1 $f(x; 1, 1)$.

## 4    Continuous function approximations by Logmoid-1

In this section the sub-family Logmoid-1 will be used to approximate any continuous function on a compact set. Logmoid-1 is a smooth and non-monotonic activation function beyond ReLU. Meanwhile, it inherits the self-gated property of Swish where the self-gated refers to the input itself and a function with the input as its argument, i.e., $f(x) = xg(x)$.

The output landscapes of a five-layer fully connected network are shown in Fig.2. While ReLU has lot of sharp transitions. Logmoid-1 shows a continuous and fluent transition shape. This is then useful for optimization and generalization (Li et al., 2018).

Next, we construct a bell-shaped function with Logmoid-1 and present main theorems for approximate functions.

**Bell-shaped functions**–From Eq.(7) Logmoid-1 with $\alpha = \beta = 1$ is defined as

$$g(x) = x \ln(1 + \sigma(x)), x \in \mathbb{R} \tag{8}$$

According to ref.(Cardaliaguet & Euvrard, 1992), define a new bell-shaped function as

$$\Psi(x) = \begin{cases} \psi(x) + 2|Y_1|, & \text{if } T_0 \leq x < T_1; \\ \lambda(\psi(x) + 2|Y_2|), & \text{if } T_2 < x < T_0; \\ -\psi(x), & \text{if } x \geq T_1; \\ -\lambda\psi(x), & \text{if } x \leq T_2. \end{cases} \tag{9}$$

where $\psi(x)$ is defined by $\psi(x) = \varphi(x + \frac{1}{2}) - \varphi(x - \frac{1}{2})$ with a bounded function $\varphi(x) = g(x + \frac{1}{2}) - g(x - \frac{1}{2})$, $\lambda = 0.934$, $T_0 = -0.2536$, $T_1 = 3.5025$ and $T_2 = -3.7326$ are extreme points, and $Y_1 = -0.0181$ and $Y_2 = -0.0307$ are the values of $\psi(x)$ at points $T_1$ and $T_2$. Here, $\lambda$ is used to ensure the continuity of $\Psi(x)$ at $T_0$.

And then, we define another bell-shaped function as

$$\Phi(x) = \frac{1}{\xi}\Psi(x) \tag{10}$$

where $\xi$ is defined by $\sum_{k=-\infty}^{\infty} \Psi(x-k) = \xi$. It implies that $\sum_{k=-\infty}^{\infty} \Phi(x-k) = 1$. We further define

$$\Gamma(x) = \begin{cases} e^{-\frac{1}{2}x - \frac{1}{2}}, & \text{if } x > 0; \\ e^{\frac{1}{2}x - \frac{1}{2}}, & \text{if } x \le 0. \end{cases}$$

**Approximation errors**–As universal approximation operators, Feed-forward neural networks (FNNs) (Hornik et al., 1989; Cybenko, 1989) with a hidden layer and $n + 1$ neurons is represented by

$$N_n(x) = \sum_{j=0}^{n} c_j \varsigma(\langle a_j \cdot x \rangle + b_j) + d_j, x \in \mathbb{R}^s, s \in \mathbb{N} \tag{11}$$

where $\varsigma$ is an activation function, $j \in [0, n]$ and $b_j \in \mathbb{R}$ are thresholds of the hidden layer, $a_j \in \mathbb{R}^s$ are the connection weights between the input and hidden layers, $c_j \in \mathbb{R}$ are the connection weights between the hidden and output layers, $d_j \in \mathbb{R}$ are the thresholds of the output layer, and $\langle \cdot \rangle$ denotes the inner product.

In general, any continuous function on a compact set can be theoretically approximated up to any precision by increasing hidden neurons (Cybenko, 1989; Hornik et al., 1990; Hornik, 1991; Funahashi, 1989; Leshno et al., 1993). We present a tighter estimation of approximation error by Logmoid-1. Let $\mathcal{C}[-1, 1]$ be the space of continuous functions on $[-1, 1]$. For any function $f \in \mathcal{C}[a, b]$, define

$$w(f, \delta) := \max_{\substack{a \le x, y \le b \\ |x-y| \le \delta}} |f(x) - f(y)|. \tag{12}$$

$f$ is $(L, \alpha)$-Lipschitz continuous with $\alpha \in (0, 1]$ (e.g., $f \in Lip(L, \alpha)$) if there is a constant $L > 0$ such that $w(f, \delta) \le L\delta^\alpha$.

For any function $f \in \mathcal{C}[-1, 1]$, we construct a FNN operator by $\Phi(x)$ as

$$G_{Logmoid-1}(f, x) := \sum_{k=-n}^{n} f(\frac{k}{n})\Phi(nx - k) \tag{13}$$

**Theorem 1** *For the operator $G_{Logmoid-1}(f, x)$ we have*

$$|f(x) - G_{Logmoid-1}(f, x)| \le w(f, \frac{1}{n^\alpha}) + (8e^{-\frac{1}{2}n^{1-\alpha}} + \frac{4}{e})\|f\|_\infty$$

*where $\|\cdot\|_\infty$ denotes the uniform norm.*

This means the approximation error of $f(x)$ by $G_{Logmoid-1}(f, x)$ can be controlled by $n$ and $\alpha$. The proof is shown in Appendix A.

Moreover, denote $\mathcal{C}_B(\mathbb{R})$ as the set of continuous and bounded function on $\mathbb{R}$. For any function $f \in \mathcal{C}_B(\mathbb{R})$ define a FNN operator as

$$\overline{G}_{Logmoid-1}(f, x) := \sum_{n=-\infty}^{\infty} f(\frac{k}{n})\Phi(nx - k). \tag{14}$$

**Theorem 2** *For the FNN operator $\overline{G}_{Logmoid-1}(f, x)$ we have*

$$\left|f(x) - \overline{G}_{Logmoid-1}(f, x)\right| \le w(f, \frac{1}{n^\alpha}) + 8e^{-\frac{1}{2}n^{1-\alpha}}\|f\|_\infty$$

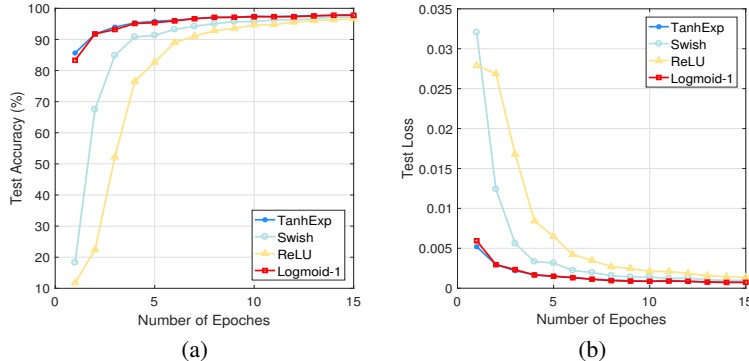

Fig. 3: Test of Logmoid-1, TanhExp, Swish, and ReLU with a 15-layer basic network on MNIST. (a) Test accuracy. (b) Test loss.

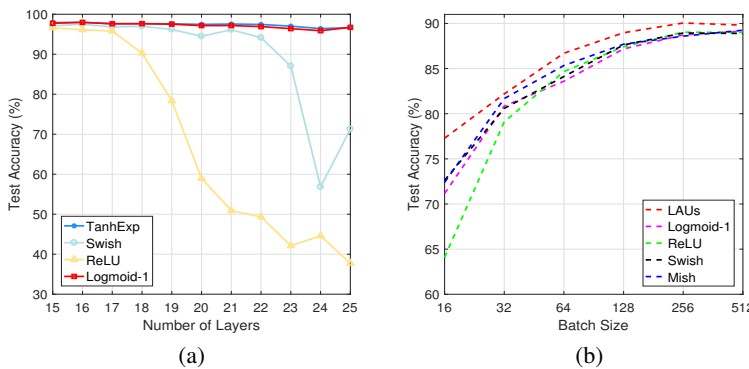

Fig. 4: (a) Test accuracy of Logmoid-1, TanhExp, Swish, and ReLU with the basic network on MNIST. (b) The test accuracy vs Batch size for LAUs, Logmoid-1, ReLU, Swish, Mish.

The proof of Theorem 2 is shown in Appendix B. Similar to the bell-shaped function (10), a new bell-shaped function based on Swish and Eqs.(9) is defined as

$$\Phi_{Swish}(x) = \begin{cases} \xi(\phi(x) + 2|Y_1|), & \text{if } -T_1 \leq x \leq T_1; \\ -\xi\phi(x), & \text{otherwise.} \end{cases} \tag{15}$$

where $\lambda = 1.758$, $T_1$ is the extreme point, and $Y_1 = -0.034 = \phi(T_1)$. $\phi(x)$ is defined by

$$\phi(x) = \text{Swish}(x+1) - 2\text{Swish}(x) + \text{Swish}(x-1) \tag{16}$$

Similar to the FNN operator $\overline{G}_{Logmoid-1}(f, x)$, we may define a FNN operator $G_{Swish}(f, x)$ by $\Phi_{Swish}(x)$, in order to further verify the performance of Logmoid-1. Two examples are shown in Appendix C.

Table 2: The Top-1 accuracy (in %) of different learning rates (lr) for LAUs, Logmoid-1, ReLU, Swish, Mish.

| lr | ReLU | Swish | Mish | Logmoid-1 | LAUs |
|----|------|-------|------|-----------|------|
| $10^{-1}$ | 87.42 | 87.62 | 87.7 | 87.18 | ●**88.97** |
| $10^{-2}$ | 87.91 | 88.15 | 88.38 | 88.36 | ●**88.58** |
| $10^{-3}$ | 77.92 | 80.38 | ●**80.99** | 80.17 | 80.44 |
| $10^{-4}$ | 50.33 | 51.48 | 53.66 | 52.53 | ●**55.36** |

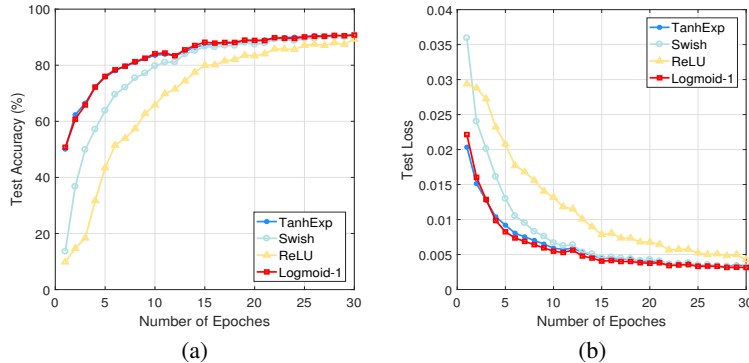

Fig. 5: Test of Logmoid-1, TanhExp, Swish, and ReLU with a 15-layer basic network on KMNIST. (a) Test accuracy. (b) Test loss.

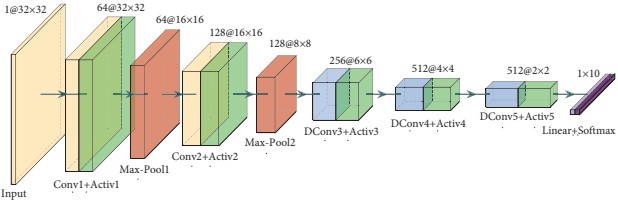

Fig. 6: Schematic VGG-8-Dilated network. DConv denotes the dilated convolutional.

## 5 EXPERIMENTS WITH LOGMOID ACTIVATION UNITS (LAU)

In this section we simulate new activation functions for different tasks. We denote Logmoid Activation Units (LAU) with parameters as

$$f(x; \alpha, \beta) = x \ln(1 + \alpha\sigma(\beta x)) \tag{17}$$

where $\alpha$ and $\beta$ are trainable hyper-parameters being optimized end-to-end to find good one at each layer automatically. Logmoid networks are feed-forward networks including convolutional and residual architectures with pooling layers. Inspired by PAUs, we suppose to learn one LAU per layer. There are $2L$ parameters with $L$ layers.

### 5.1 INITIALIZING EXPERIMENTS WITH LOGMOID-1

This subsection is for exploring good initializations for all LAUs. We show that Logmoid-1 is good beyond random initializations. Several ablation experiments are performed to evaluate Logmoid-1 on MNIST (LeCun et al., 2010) and KMNIST (Clanuwat et al., 2018) with 10 classes. All the simulations used a basic network with 15 layers (Liu & Di, 2020). Each one of the last 12 layers contains a batch normalization (Ioffe & Szegedy, 2015), a dropout rate (Srivastava et al., 2014) of 0.25, an activation function, and a dense layer with 500 neurons.

**Learning speed**. We trained the basic 15 layers network with 15 epoches on MNIST and 30 epoches on KMNIST. As shown in Figs.3 and 5, Logmoid-1 performs better than Swish and ReLU both in the convergence speed and final accuracy, and has a similar performance with TanhExp. Logmoid-1 can update the parameters rapidly and force the network to fit the dataset in an efficient way.

**Over-fitting**. Experiments are performed with Logmoid-1, ReLU, Swish, and TanhExp on both MNIST and KMNIST using the basic network with blocks from 15 to 25. From Fig.4(a), the network suffers from over-fitting by ReLU and Swish because of a sharp decrease in accuracy. Logmoid-1 and TanhExp maintain a high accuracy in large models and then prevent from over-fitting.

## 5.2 Experiments with LAUs

This subsection devotes to the simulation experiments with LAUs for different tasks. We simulate LAUs and other baseline functions including ReLU, Swish, TanhExp, ACONC (Ma et al., 2021) and Logmoid-1 using standard deep neural networks. We benchmark the results in the following datasets: MNIST (LeCun et al., 2010), Kuzushiji-MNIST (KMNIST) (Clanuwat et al., 2018), Fashion-MNIST (FMNIST) (Xiao et al., 2017), CIFAR (Krizhevsky et al., 2009), ImageNet (Russakovsky et al., 2015) and COCO (Lin et al., 2014).

**Ablation experiments with LAUs**–We demonstrate the image classification performance with LAUs and others. ShuffleNet-V2 network is used as the backbone. All the experiments are simulated on CIFAR-10 dataset.

*Batch size*. We test LAUs in the same network with the batch size of 16, 32, 64, 128, 256, 512, respectively. LAUs has the best performance compared with others as Fig.4(b).

*Learning rate*. Table 2 shows the Top-1 accuracy in ShuffleNet-V2 network. For the learning rate $10^{-1}$, LAUs has the highest 88.97% test accuracy while three nonmonotonic functions have similar performance with $10^{-2}$. LAUs is best for learning rate $10^{-4}$. This means LAUs has good performance in almost all comparable learning rates. Meanwhile, its expression ability is good at a smaller learning rate.

Table 3: Performance comparison of activation functions on FMNIST.

| Activation | LeNet(#0.06M) | VGG-8(#9.2M) | VGG-8-Dilated(#3.9M) |
|---|---|---|---|
| ReLU | ∘89.84 | 90.19 | 90.96 |
| Swish | 89.37 | 90.15 | 90.80 |
| TanhExp | 89.47 | 90.72 | 90.45 |
| ACONC | 89.27 | ∘91.23 | ∘91.12 |
| Lgomoid-1 | 88.99 | 90.56 | 90.25 |
| LAUs | **●89.90** | **●92.35** | **●91.84** |

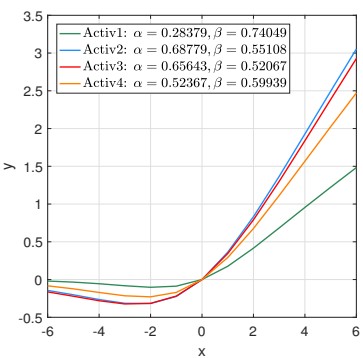

Fig. 7: The estimated activation functions after training LeNet with LAUs on FMNIST.

**Empirical results on Fashion-MNIST**–We take use of two basic architectures of LeNet (Lecun et al., 1998) and VGG-8 (Simonyan & Zisserman, 2014) to evaluate Logmoid-1 on FMNIST. LeNet contains 61,706 trainable parameters while LAUs adds 8 additional parameters. We investigate whether there are smaller sub-networks with similar performances. Some blocks of VGG-8 will be replaced by dilated convolutions, as Fig.6, named as VGG-8-Dilated network.

From Table 3, LAUs improves 1.1% Top-1 accuracy compared with the runner-up result on VGG-8. Comparing the baseline functions on the VGG-8-Dilated and LeNet, LAUs always matches or outperforms the best performance. This shows LAU is a learnable activation function without requirement of additional suboptimal experiments. From Fig.7 some learnable activations are

smooth versions of Logmoid-1 by controlling hyper-parameters $\alpha$ and $\beta$. So, we initialize LAUs with Logmoid-1.

**Empirical results on CIFAR**–We explore the performance of LAUs on difficult datasets, CIFAR-10 and CIFAR-100 (Krizhevsky et al., 2009). These 3-channel images require a neuron network with strong learning ability. We simulate LAUs as well as other baseline functions on ResNet-20 (He et al., 2016), (Howard et al., 2017), MobileNet-V2 (Sandler et al., 2018), ShuffleNet (Zhang et al., 2018), ShuffleNet-V2 (Zhang et al., 2018), SqueezeNet (Iandola et al., 2016), SeNet-18 (Hu et al., 2018) and EfficientNet-B0 (Tan & Le, 2019).

From Table 4 LAUs outperforms the baseline activation functions in most cases. LAUs improves the Top-1 accuracy by around 5% on EfficientNet-B0. The predictive performance of LAUs on ResNet-20 and SqueezeNet is still competitive. LAUs gets 2.49% improvement over ReLU on MobileNet while SRS (Zhou et al., 2020) gets 2.33% improvement.

Table 4: Performances of activation functions on CIFAR-10.

| Model | ReLU | Swish | ACONC | TanhExp | Logmoid-1 | LAUs |
|---|---|---|---|---|---|---|
| ResNet-20 | 90.96 | 91.08 | ∘91.16 | •91.26 | 90.97 | ∘91.12 |
| MobileNet | 86.04 | 87.93 | ∘88.31 | 88.24 | 87.91 | •88.53 |
| MobileNet-V2 | 84.92 | 85.87 | ∘86.54 | 85.83 | 86.39 | •87.95 |
| ShuffleNet | 87.55 | 88.87 | 88.95 | ∘89.2 | 87.22 | •89.91 |
| ShuffleNet-V2 | 87.42 | 87.62 | ∘87.64 | 87.38 | 87.18 | •88.97 |
| SqueezeNet | •88.50 | 87.04 | ∘88.35 | 87.86 | 87.63 | ∘88.22 |
| SeNet-18 | 88.81 | 90.53 | 90.37 | 90.19 | ∘90.95 | •91.43 |
| EfficientNet-B0 | 76.17 | 77.35 | ∘78.55 | 77.19 | 78.23 | •82.52 |

Table 5: Performances of activation functions on CIFAR-100.

| Model | ReLU | Swish | ACONC | TanhExp | Logmoid-1 | LAUs |
|---|---|---|---|---|---|---|
| ResNet-20 | 66.16 | 67.73 | •67.86 | ∘67.79 | 67.68 | ∘67.77 |
| MobileNet | 59.40 | 62.11 | •62.35 | ∘62.21 | 61.81 | ∘62.05 |
| MobileNet-V2 | 59.40 | 61.32 | ∘62.13 | 60.76 | 60.09 | •63.66 |
| ShuffleNet | 64.53 | 65.45 | 66.17 | 66.7 | ∘66.76 | •68.01 |
| ShuffleNet-V2 | 63.65 | 65.88 | ∘65.94 | •66.18 | 65.69 | 64.73 |
| SqueezeNet | 62.64 | ∘63.97 | 63.45 | 62.38 | 63.64 | •64.26 |
| SeNet-18 | 66.26 | ∘67.85 | 67.78 | 67.68 | 67.45 | •68.96 |
| EfficientNet-B0 | 45.81 | 46.01 | ∘48.26 | 47.52 | 47.08 | •50.38 |

The CIFAR-100 has 100 classes with 500 training images and 100 test images per class. From Table 5 LAUs outperforms ReLU on all networks. It improves the top-1 accuracy by 4.6%, 3.3%, 4.4%, 2.1% and 2.9% compared with ReLU, Logmoid-1, Swish, ACONC and TanhExp on EfficientNet-B0. LAUs gets 2.65% improvement over ReLU on MobileNet and is smaller than its from SRS. LAUs shows smaller classification accuracy by 1.4% compared with the best on ShuffleNet-V2. This may be further improved by increasing simulation epochs.

Table 6: Comparison of different activations on the ImageNet-200 dataset. We report the Top-1 and Top-5 accuracies (in %) on ShuffleNet-V2 and MobileNet-V3 (Howard et al., 2019).

| | ReLU | | Swish | | ACONC | | TanhExp | | Logmoid-1 | | LAUs | |
|---|---|---|---|---|---|---|---|---|---|---|---|---|
| | Top-1 | Top-5 | Top-1 | Top-5 | Top-1 | Top-5 | Top-1 | Top-5 | Top-1 | Top-5-1 | Top-1 | Top-5 |
| ShuffleNet-V2 | 59.82 | 82.21 | 60.38 | 83.15 | 62.35 | 84.68 | 60.05 | 83.07 | ∘63.07 | ∘85.02 | •63.83 | •86.03 |
| MobileNet-V3 | 45.56 | 72.15 | 46.49 | 73.88 | 47.52 | 74.79 | 46.21 | 73.36 | ∘48.65 | ∘76.15 | •50.16 | •77.35 |

**Empirical results on ImageNet-200**–We compared LAUs with other baseline activation functions on ImageNet 2012 (Russakovsky et al., 2015). For a quick comparison, we randomly choose 200 classes and extracted 500 training images, 50 testing images each class from ImageNet, named as ImageNet-200. From Table 6 LAUs leads in both the Top-1 and Top-5 accuracies. LAUs gets 4.6% Top-1 improvements over ReLU on MobileNet-V3 while PWLU (Zhou et al., 2021) gets 1.91% Top-1 improvements.

Table 7: Performances of activation functions on the COCO object detection task. We report results on Mask R-CNN with Swin Transformer backbone.

| Activation | $AP^{box}$ | $AP^{box}_{50}$ | $AP^{box}_{75}$ | $AP^{mask}$ | $AP^{mask}_{50}$ | $AP^{mask}_{75}$ |
|---|---|---|---|---|---|---|
| ReLU | 45.8 | 64.6 | 50.3 | 41.2 | 63.3 | 44.7 |
| ACONC | 47.4 | 66.3 | 50.9 | 42.4 | 65.2 | 45.5 |
| LAUs | **48.7** | **67.5** | **52.1** | **43.5** | **66.8** | **46.9** |

**Object detection on COCO**–Object detection is a fundamental branch of computer vision. We implement simulation experiments on COCO 2017 (Lin et al., 2014), which contains 118K training, 5K validation and 20K test-dev images. We choose the Mask R-CNN (He et al., 2017) as the detector and Swin Transformer (Liu et al., 2021) with different activations as the backbone. We choose a batch size of 2, AdamW optimizer (initial learning rate of 0.0001, weight decay of 0.05, and 3x schedual). From Table 7, LAUs gets $2.9\% AP^{box}$ and $2.3\% AP^{mask}$ improvements respectively over ReLU while PWLU (Zhou et al., 2021) gets $1.42\%\ AP^{box}$ and $1.83\% AP^{mask}$ improvements.

## 6  CONCLUSIONS

In this work, we presented a novel family of nonmonotonic activation function, named as Logmoid family. We constructed a class of FNN operators with Logmoid-1 to approximate any continuous function. We proposed a learnable Logmoid Activation Unit (LAU), which is initialized using Logmoid-1, trainable in an end-to-end fashion. One can replace standard activation functions with LAU units in any neural networks. Simulations shows LAUs has the best performance across all activation functions and architectures. Further work will be performed to apply LAUs to other related tasks.

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
