# OpenReview forum: "LAU: A novel two-parameter learnable Logmoid Activation Unit"
_ICLR.cc/2023/Conference — Submitted to ICLR 2023_

### Official Review · Reviewer_KSN3 · 2022-10-15

**Confidence:** 4
**Clarity, Quality, Novelty And Reproducibility:** Exceeds page limit.
**Correctness:** 1
**Technical Novelty And Significance:** 1
**Empirical Novelty And Significance:** Not applicable
**Recommendation:** 1

**Strength And Weaknesses:**

Exceeds page limit.

**Summary Of The Paper:**

This paper is 16 pages (+references), while the strict upper bound for ICLR is 9 pages. Thus I treat this paper as desk rejected.

**Summary Of The Review:**

Exceeds page limit, desk reject.

---

> ### Author Response · Authors · 2022-11-17
> **Reply to comments**
>
> This paper is 16 pages (+references), while the strict upper bound for ICLR is 9 pages. Thus I treat this paper as desk rejected.
>
> Reply: Thank you for comments. We have revised it into a new one with 9 pages in main texts.
>
>
> Correctness: 1: The main claims of the paper are incorrect or not at all supported by theory or empirical results.
>
> Reply: Thank you for comments. We believe the main clains are correct and supported by both theory and  empirical results. In fact,  the theory results are shown in Theorems 1 and 2 by strict proofs. Both provide theoretical results for approximating any continous functions by the present activation functions. The simulations are shown in Sec.5. We have tested it for lots of datasets for different classification tasks, as shown in Tables 2-7. All of these results have shown its good performances beyond most of well-known functions. So, the present results are well-supported by both theory and empirical results.
>
> Thank you for great comments.
>
> Best.

---

### Official Review · Reviewer_Ea8B · 2022-10-24

**Confidence:** 5
**Correctness:** 2
**Technical Novelty And Significance:** 2
**Empirical Novelty And Significance:** 2
**Recommendation:** 3

**Clarity, Quality, Novelty And Reproducibility:**

Clarity: bad
Quality bad
Novelty: limited
Reproducibility: best


**Strength And Weaknesses:**

Strength:
The proposed LAUs are simple and easy to implement.

Weakness:
1.	Insufficient experiments. Pade is introduced in Section Related Works, but the authors did not compare LAUs with Pade among all the benchmark experiments. Besides, there are too many experiments on tiny datasets and networks, making the improvements not very impressive and convincing. Similar to Pade, I suggest the authors provide results of LAUs on ImageNet-1K with MobileNet-V2. Furthermore, there are many powerful activation functions like ACON (Activate or Not: Learning Customized Activation) and PWLU (Learning specialized activation functions with the Piecewise Linear Unit) that the authors did not see. To make the results more convincing, I suggest authors compare LAUs with ACON and PWLU.
2.	This paper is badly written. Section 4.1 suddenly appears without any introduction, which makes readers confused. And Section 4.2 is unnecessary because Section 5 provides experiments about Logmoid-1. All the tables are not centered. Table and figure captions are not clear enough for readers to understand.
3.	Limited novelty. There are many Sigmoid-based activation functions like Soft-Root-Sign and ACON, and it is convenient to provide different Sigmoid-based activation functions with search methods. So I do not see much novelty in this paper.


**Summary Of The Paper:**

This paper proposes a new family of activation functions called LAUs (Logmoid Activation Units). LAUs have two parameters α and β, and the authors comprehensively discuss the change of LAUs and their derivatives with different α and β. The authors also prove that Logmoid-1 (a special case of LAUs) makes Feed-forward neural network achieves lower approximation error than Swish with both mathematical and experimental results. The authors show that LAUs perform better than baseline activation functions on different datasets and architectures.

**Summary Of The Review:**

The proposed LAUs are Sigmoid-based activation functions. The overall novelty is limited, and missed essential experiments. The paper is bad-written with the problem of duplication and lack of logical coherence.

---

> ### Author Response · Authors · 2022-11-17
> **Reply to comments**
>
> Strength And Weaknesses:
> Thank you for comments. We have revised according your comments.
>
> Weakness: 1. Insufficient experiments. Pade is introduced in Section Related Works, but the authors did not compare LAUs with Pade among all the benchmark experiments. Besides, there are too many experiments on tiny datasets and networks, making the improvements not very impressive and convincing. Similar to Pade, I suggest the authors provide results of LAUs on ImageNet-1K with MobileNet-V2. Furthermore, there are many powerful activation functions like ACON (Activate or Not: Learning Customized Activation) and PWLU (Learning specialized activation functions with the Piecewise Linear Unit) that the authors did not see. To make the results more convincing, I suggest authors compare LAUs with ACON and PWLU.
>
> Reply: Thank you for suggestions. We add new experiments on MobileNet-V2 and COCO. The present LAUs is good than ACON shown in Tables 3-7. From Table 7, LAUs gets 2:9%APbox and 2:3%APmask improvements respectively over ReLU while PWLU gets 1.42% APbox and 1:83%APmask improvements. This provides a quick comparsions with PWLU.
>
> 2. This paper is badly written. Section 4.1 suddenly appears without any introduction, which makes readers confused. And Section 4.2 is unnecessary because Section 5 provides experiments about Logmoid-1.
>
> Reply:  Thank you for comments. We have revised it. Section 4 is for approximating any continous functions with LOGMOID-1 in theory.  While all the experiments are shown in Secion 5.
>
>  All the tables are not centered. Table and figure captions are not clear enough for readers to understand.
> Reply: Thank you for comments.  We have revised in this version.
>
> 3. Limited novelty. There are many Sigmoid-based activation functions like Soft-Root-Sign and ACON, and it is convenient to provide different Sigmoid-based activation functions with search methods. So I do not see much novelty in this paper.
>
> Reply: Thank you for comments. Although there are many Sigmoid-based activation functions, no one performs best for all tasks. Our goal in this paper is for a novel learnable Logmoid Activation Unit (LAU) by parameterizing Logmoid with two hyperparameters that are optimized via back-propagation algorithm. The limited hyperparameters provide great improvements than most of well-known functions including Sigmoid, Swish, and ReLU. This provides a simple way for improving experimental performances in various classification tasks. We believe it should be interesting for verious experts, especially for middle-size networks in hand-by computers.
>
> Thank you for your great comments.
> Best.

---

### Official Review · Reviewer_MpkZ · 2022-10-25

**Confidence:** 4
**Correctness:** 4
**Technical Novelty And Significance:** 3
**Empirical Novelty And Significance:** 2
**Recommendation:** 5

**Clarity, Quality, Novelty And Reproducibility:**

I like the idea and the theoretical analysis, but not satisfied with the experiments. A good activation function should be able to work well for multi-tasks, multi-architectures and large scale dataset, which is not validated in the paper.

**Strength And Weaknesses:**

Strength:
1. an interesting activation function with two parameters that covers the geometry of existed activation functions.

2. theoretical analysis in terms of approximating continuous functions.

3. excellent results achieved over multiple network structures and datasets.

Weakness:
the experiments are far from extensive.
1. we do not know if this activation works for other tasks like detection/segmentation.
2. it is unknown if this activation works for transformer.
3. it is unknown if this activation works well with dynamic operators like SE in MobileNet-V3.
4. it is unknown if this activation works well for large scale dataset (ImageNet-1K, ImageNet-22K).


**Summary Of The Paper:**

This paper introduces a new logmoid activation unit, that has nice properties to approximate continuous function and introduce performance gain over other activation functions across multiple datasets.

**Summary Of The Review:**

This paper is borderline: interesting idea, good analysis, but weak experiments (see weakness)

---

> ### Author Response · Authors · 2022-11-17
> **Reply to comments**
>
> Thank you for great comments. We have revised it according your suggestions.
>
> Weakness: the experiments are far from extensive.
>
> 1. we do not know if this activation works for other tasks like detection/segmentation.
>
> Reply: In Section 5, we add object detection on COCO over ReLU and ACONC, and our activation works better.
>
> 2. It is unknown if this activation works for transformer.
>
> Reply: Thank you for suggestions.  In Section 5, we have carried out object detection experiments on the currently popular Swin Transformer framework for computer image processing, experiments show that our activation still has a big advantage.
>
> 3. It is unknown if this activation works well with dynamic operators like SE in MobileNet-V3.
>
> Reply:Thank you for suggestions.  In Section 5,  we replace the activation in Mobilenet-V3 including the SE module by LAUs,  and get a better result on Imagenet-200.
>
> 4. It is unknown if this activation works well for large scale dataset (ImageNet-1K, ImageNet-22K).
>
> Reply: Thank you for suggestions. Due to limited time we could not fully test the ImageNet-1k or Imagenet-22k datasets. Instead,  in section 5, we randomly selected 200 classes on imageNet-1k and retained the size of the original image. This simulation results show good performance by LAUs. We believe similar result may be obtained for other classes. This provides a quick comparsion of different activation function on large scale dataset.
>
> Clarity, Quality, Novelty And Reproducibility:
> I like the idea and the theoretical analysis, but not satisfied with the experiments. A good activation function should be able to work well for multi-tasks, multi-architectures and large scale dataset, which is not validated in the paper.
>
> Reply:In the revision, we have added some new simulations for multi-tasks, multi-architectures and large scale dataset (in random). All the results show the good performances by LAUs beyond other activations. We hope it might meet the requirement.
>
> Thank you for great comments.
>
> Best.

---

> > ### Comment · Reviewer_MpkZ · 2022-12-02
> > **Thanks for the reply**
> >
> > Thank authors for the reply, including more explanation and experiments. This definitely improves the paper quality. I would suggest authors finish experiments on ImageNet-1K (optional for ImageNet-22K), as ImageNet-1K is a standard dataset for most of works on architectures or operators. Another important properties to explore is scaling from very efficient network (like TinyNet, MicroNet) to very large network (like ViT-Huge). Solid experiments over multiple dimensions (tasks, architectures, scales) become more crucial for new operators than a couple years ago.

---

### Decision · Program_Chairs · 2023-01-20

**Decision:**

Reject

**Justification For Why Not Higher Score:**

The paper is badly written and the reviewers unanimously agreed to reject this paper. The initial submission was significantly over the page limits.

**Justification For Why Not Lower Score:**

There is no lower score.

**Metareview: Summary, Strengths And Weaknesses:**

## Summary

This paper proposes a novel activation unit for neural networks called LAU with parametrization of $y= x \text{ln}(1+\alpha \text{sigmoid}(\beta x))$ where $\alpha$ and $\beta$ are learnable parameters that are optimized with backpropagation. This gives neural networks the ability to approximate any continuous activation function in closed spaces. The paper presents empirical evidence that the LAU outperforms ReLU on the benchmark datasets they proposed.

The authors agreed to reject this paper. The initial submission of the paper was over the page limit and it was 19 pages which was a violation of the submission guidelines. As it stands this paper is not publication-ready yet.

Below I will summarize some of the strengths and weaknesses pointed by the other reviewers.

## Strengths

- Strong experimental results.
- Simple and easy to implement.
- Theoretical results.

## Weaknesses

- Experiments are limited to classification tasks only. It is not clear if they would work for detection or segmentation only.
- The experiments are only limited to typical feedforward architectures used for classification. It is not clear if they would work with transformers or LSTMs, for instance.
- The experiments focus on small-scale datasets. It is not clear if the results would scale well to the larger datasets and larger models.
- The paper is badly written.
- Initial submission is significantly over page limit.